# Deep Learning-Based Prediction of Molecular Tumor Biomarkers from H&E: A Practical Review

**DOI:** 10.3390/jpm12122022

**Published:** 2022-12-07

**Authors:** Heather D. Couture

**Affiliations:** Pixel Scientia Labs, Raleigh, NC 27610, USA; heather@pixelscientia.com

**Keywords:** deep learning, digital pathology, molecular tests, precision medicine, biomarkers

## Abstract

Molecular and genomic properties are critical in selecting cancer treatments to target individual tumors, particularly for immunotherapy. However, the methods to assess such properties are expensive, time-consuming, and often not routinely performed. Applying machine learning to H&E images can provide a more cost-effective screening method. Dozens of studies over the last few years have demonstrated that a variety of molecular biomarkers can be predicted from H&E alone using the advancements of deep learning: molecular alterations, genomic subtypes, protein biomarkers, and even the presence of viruses. This article reviews the diverse applications across cancer types and the methodology to train and validate these models on whole slide images. From bottom-up to pathologist-driven to hybrid approaches, the leading trends include a variety of weakly supervised deep learning-based approaches, as well as mechanisms for training strongly supervised models in select situations. While results of these algorithms look promising, some challenges still persist, including small training sets, rigorous validation, and model explainability. Biomarker prediction models may yield a screening method to determine when to run molecular tests or an alternative when molecular tests are not possible. They also create new opportunities in quantifying intratumoral heterogeneity and predicting patient outcomes.

## 1. Introduction

Breast cancer is a clear example of the effectiveness of precision medicine. Tumors that overexpress the HER2 protein can receive targeted therapy, radically improving the prognosis [1,2,3]. Many cancer treatments are only effective for specific mutations or genomic profiles. Understanding the subset of patients who may benefit is the key to personalized cancer treatments. Patients can be stratified by a variety of molecular and genomic properties, where each subtype responds to some courses of treatment but not others and tends to have a distinct prognosis. Advances in precision medicine over the last decade have focused on these molecular biomarkers, which have helped identify the significant proportion of patients who don’t respond to immunotherapy.

The technologies used in assessing these molecular properties are expensive and time-consuming to perform. They may involve DNA sequencing to detect mutations, RNA sequencing to assess gene expression, or immunohistochemical staining to identify protein biomarkers. Most are not routinely performed on all patients who could benefit and are not done at all in labs with limited resources. To complicate things further, in many cases only a small amount of tissue is excised from a tumor, and there is not enough for additional analyses beyond what a pathologist examines through the microscope. New studies keep identifying more molecular properties of potential clinical value, each requiring its own tissue slice and processing procedure. Current workflows are not designed to incorporate this many tests.

While comprehensive molecular testing will be difficult to implement at scale, histological staining of tissue is common practice and imaging of such samples has become increasingly available with the transition to digital pathology. Recent advances in computational pathology have made a new alternative possible: machine learning algorithms applied to hematoxylin and eosin (H&E) histopathology. Dozens of studies—all published in the last few years—have demonstrated that one or more molecular properties can be predicted from H&E alone using the advancements of deep learning.

Correlation between morphological features and certain genomic properties was already established prior to the recent popularization of deep learning. Colorectal cancer microsatellite instability (MSI) can be predicted by tissue characteristics like the presence of tumor infiltrating lymphocytes [4,5]. However, the accuracy of such a model is limited and may have even less differentiating power for other biomarkers. The earliest machine learning approaches to predicting molecular biomarkers from H&E used classical computer vision features like key point descriptors [6] or dictionary learning [7]. These studies showed limited success; more powerful features are needed to accurately capture complex histological signatures.

Recent advances in deep learning can now decipher patterns that are beyond the limits of human perception. Four types of molecular biomarkers have been successfully predicted from H&E in studies of more than ten different types of cancer: (1) protein biomarkers, (2) genomic subtypes and expression of individual genes, (3) molecular alterations, (4) virus. Previous review articles have focused more broadly on artificial intelligence in digital pathology [8,9], on select subsets of biomarkers (e.g., predicting MSI [10,11,12,13]), or on the types of molecular features that can be predicted [14,15]. Similar machine learning methods apply across these biomarkers, and this review will emphasize the machine learning algorithms that enable these advancements in precision medicine. Section 2 will first examine the types of models that have been developed to predict molecular biomarkers from H&E whole slide images (WSIs). The challenges in developing this type of model will be discussed in Section 3. Finally, Section 4 will look at some opportunities for further advancing this area of research.

## 2. Deep Learning Methods for Whole Slide Images

Traditional methods in computational pathology mostly focus on replicating tissue morphology features to mimic pathologists. Advancements in deep learning over the last decade have brought feature learning methods to the forefront. The majority of the recent success in predicting biomarkers is attributed to deep learning. I refer to these methods as bottom-up approaches, as they learn important patterns directly from the pixels. This section will first look at bottom-up approaches, followed by pathologist-driven and hybrid models, and, finally, strongly supervised models for when a spatially-localized ground truth is available. Figure 1 provides an overview of these four frameworks.

### 2.1. Bottom-Up Approaches to Learning Features

Most early applications of machine learning to histopathology focused on classifying or segmenting image tiles. These approaches require detailed annotations at the tile or pixel level. For classifying tissue types or segmenting nuclei, annotations are time-consuming but possible. Molecular biomarkers, on the other hand, are usually only available at the patient level. Typically there is no way for a pathologist to know which regions of tumor are associated with a mutation or genomic subtype. For this reason, most algorithms to predict molecular biomarkers rely on weak supervision in which only a patient-level label is available for training.

#### 2.1.1. Learning Tile Features

Weak supervision makes it challenging to learn a representation for image tiles. Most solutions have relied upon transfer learning with a convolutional neural network (CNN) pretrained on the ubiquitous dataset of computer vision: ImageNet. While the features are not tuned to histopathology images, they are still quite transferable—particularly the lower CNN layers. Many studies have taken this efficient approach [16,17,18,19], while others finetuned the CNN by applying the patient-level labels to each tile in a fully supervised setup [20,21,22,23,24,25,26]. Recent advances in self-supervised learning present another alternative: pretrain a model on tiles from WSIs with a pretext task in order to learn a better representation for histopathology. A detailed review of pretext tasks used for pathology goes beyond the scope of this article; see [27,28,29] for a good overview on histopathology.

One example of self-supervised learning is Rawat et al.’s image representation called “fingerprinting” that learns features to distinguish one patient from another [30]. They demonstrated that this representation works better than a fully supervised approach for predicting breast cancer receptor status. Self-supervision can also be used jointly with a fully supervised task. Liu et al. proposed a meta contrastive learning framework that iteratively trains a model with cross-entropy and contrastive objectives [31]. The contrastive objective increases intra-class similarity and decreases inter-class similarity.

Transformers have also emerged as an alternative to the CNN. Guo et al. trained a Swin-T transformer for predicting MSI and a few other colorectal cancer biomarkers [32]. Their model produced results superior to previously published studies and was significantly more robust than other methods when trained on 50% or 25% of the training set. Transformers are still new to computer vision and even newer to histopathology. There will be many more developments to come.

#### 2.1.2. Tile Aggregation

The first step in processing WSIs is selecting which tiles to use for model training. WSIs can be as large as 100,000×100,000 pixels. Tiles containing little tissue can easily be discarded by tissue detection methods, and most studies also exclude non-tumor tissue. The simplest training setup is a fully supervised model in which each tile is assigned the patient-level label, for example, positive or negative for a particular biomarker. Then a CNN is trained using the set of tiles and their corresponding label. At inference time, tile predictions are combined into a patient-level prediction.

The most common method is a simple aggregation of tile predictions using the mean prediction or taking a majority vote [20,25,33,34,35,36,37]. This approach will work well in some cases, but may fail if the tumor is heterogeneous or not all tiles are informative of the biomarker status—for example, if non-tumor tiles are included. Somewhat more robust strategies involve a second stage classifier to make patient-level predictions. For example, La Barbera et al. calculated statistics on the tile predictions to form a single patient-level feature vector on which the second stage classifier was trained to predict HER2 status of breast cancer [38]. Alternatively, a quantile function can represent the distribution of tile predictions [17]. A second classifier is then trained using the quantile function for each tumor and used to predict the final class. Quantile functions can also be incorporated into a CNN as a pooling layer so that the whole model may be trained end-to-end [39].

Rather than training a tile classifier and a second stage model to aggregate tile predictions, most research now favors a single model to aggregate tile features. Recurrent neural networks (RNNs) have been used to make a prediction after encoding a sequence of tiles [40]. More frequently, a multiple instance learning (MIL) model is chosen to accommodate the latent tile labels. MIL models handle the weak supervision aspect of patient-level labels; the tile labels are unknown and must be inferred by the algorithm. Some studies have experimented with methods like CHOWDER that focus on ranking the tiles, then make predictions based on the top and possibly bottom *N* tiles [41,42]. However, self-attention is more frequently chosen due to its superior results [19,31,41,43,44,45,46,47,48,49]. Following DeepMIL by Ilse et al., an attention model combines tile features as a weighted sum, where the attention weights are learned by the model itself from the tile features [50]. This way the model can place more weight on tiles that are informative for the specific task and less on non-discriminative tiles. Figure 2 demonstrates inference with an attention model.

An attention model can be applied as a CNN aggregation layer for end-to-end prediction. In order to fit the entire model on a GPU for training, a subset of tiles must be selected from each WSI. For example, Hohne et al. selected thirty tiles from each WSI of thyroid cancer, with this selection made randomly for each epoch while training to predict mutation status [43]. Alternatively, each tile may be encoded by a pretrained model, compressing it down to a smaller representation [41,44,45,51]. This way, the encoded features for the entire WSI may fit on the GPU, and the attention model is learned using all tiles.

In comparing the performance of an attention-based model to simpler alternatives, research has largely shown attention models to be superior [41,45,51]. One exception is a study by Leleh et al. that compared four MIL models: a naive model that applied the patient label to each tile and finetuned the model, a classical MIL model that used max-pooling to select the highest prediction for a slide, an attention model, and a clustering-constrained attention model [52]. They found that the non-MIL models were usually superior; however, they only finetuned the ImageNet-pretrained CNN for the naive model, making this an unfair comparison with MIL approaches. Weitz et al. investigated the impact of noisy instances on MIL models and found that, while attention models performed well on the internal test set, a simple mean of predictions was better on an external cohort [53]. However, they also used an ImageNet-pretrained model without finetuning. The type of data that the CNN was tuned on can result in quite different conclusions. In particular, the use self-supervised learning can lead to different conclusions about MIL [41,45].

#### 2.1.3. Weakly Supervised Learning with Self-Supervised Features

Self-supervised features can be used in any of the aggregation models discussed in Section 2.1.2. However, the most frequent pairing is with an attention model. Abbasi-Sureshjani et al. used attention-based deep MIL with a backbone CNN pretrained in one of two ways: on ImageNet or on histopathology images with the contrastive learning method BYOL [44]. In most cases, the two backbones performed similarly for breast cancer molecular subtypes. However, the self-supervised one was significantly better when the model was tested on slides from a different scanner than the model was trained on. An additional example is a study in which Saillard et al. compared three different MIL frameworks for predicting MSI using either an ImageNet-pretrained CNN or a contrastive one using the method MoCo [41]. The best MIL framework varied somewhat with the dataset; however, the self-supervised variants consistently produced better results than the ImageNet ones, especially when applying the model to a different cohort. The self-supervised models also provided a clearer interpretation for pathologists of the patterns learned by the model. The improved generalization performance in these two studies demonstrates a powerful potential for self-supervised learning.

While most tumors likely belong to a single class, others may be heterogeneous. Some portions may have a particular mutation while the rest does not, or more than one molecular subtype can be present. Schirris et al. took the DeepMIL framework a step further to incorporate a measure of heterogeneity [45]. They pretrained a CNN using the self-supervised learning method SimCLR, then used it to encode tiles from WSIs. The tile representations are combined with a deep MIL model that incorporates an attention module like DeepMIL and also integrates a measure of feature variance to capture heterogeneity. Once again, self-supervised pretraining produced a significant improvement in model performance compared to a model pretrained on ImageNet. The attention-based multiple instance component also improved performance significantly with an additional small boost from attention-weighted variance.

#### 2.1.4. Histopathology-Based Transfer Learning

Self-supervised learning is a powerful method for pretraining a CNN. An alternative means of transfer learning that is particularly relevant to the topic of this article is pretraining to predict a biomarker. Schmauch et al. pretrained their model by predicting RNA-Seq gene expression [54]. They first trained their HE2RNA model to predict gene expression. Predictions for some genes were more accurate than others, and this model alone produced some fascinating results. Then they transferred the gene expression model to a much smaller dataset to predict MSI. Other studies have also predicted gene expression directly. Levy-Jurgenson et al. trained models to predict mRNA and miRNA expression for breast and lung cancer [55], while Wang et al. studied mRNA expression for more than seventeen thousand genes in breast cancer [56]. Gathering a sufficiently large histopathology dataset for training a deep learning model is challenging. Gene expression models could provide a means to pretrain on a large dataset like The Cancer Genome Atlas (TCGA) and finetune on a dataset with fewer patients.

### 2.2. Learning with Pathologist-Driven Features

The above methods all focus on learning patterns directly from images using deep learning. The alternative is models using hand-crafted features that are typically more interpretable.

#### 2.2.1. Hand-Crafted Tissue Features

Prior to the development of deep learning, hand-crafted features mostly consisted of image processing techniques to characterize tissue and cell morphology. Methods like segmenting individual nuclei and representing their size, shape, intensity, texture, and spatial arrangement. Chauhan et al. extracted these types of features and summarized each over the tile using the mean, standard deviation, entropy, skewness, and kurtosis [57]. They found that the most important features varied with the biomarker; however, morphology and intensity were generally the most critical, with a slight improvement from spatial features for some biomarkers.

Deep learning now frequently plays a role in creating pathologist-driven features. Sadhwani et al. used established histologic patterns to characterize tissue [58]. They segmented tissue into nine different histologic classes to create interpretable features and compared with a weakly supervised CNN for predicting tumor mutational burden of lung cancer. They also tried combining each with clinical variables, as well as a hybrid model that included all features. The hybrid model performed best, but some of the other variations were not far behind. These results demonstrate the potential of interpretable methods for tissue classification. Diao et al. also focused on human-interpretable features [59]. They classified tissue and cell types with deep learning, then extracted features from each. These features were then aggregated across many images and mapped to biological concepts such as the PD-1 biomarker.

#### 2.2.2. Hybrid Models

The pathologist-driven methods above open the avenue for modeling the tissue microenvironment as an alternative-and more interpretable-means to predict molecular properties from WSIs. Alghamdi et al. combined the classification of cells with modeling their spatial arrangement using deep learning [60]. They detected and classified all cells on the slide into five types. Then they formed a cell map for each type that identified the location of each cell. Compressing the cell maps by applying an averaging filter produced a smaller representation with each pixel representing the density of cells in that region. Finally, they trained a CNN on these 5D cell maps to predict receptor status. Experiments demonstrated that their model significantly outperformed one trained on raw H&E image tiles.

An alternative method for mapping cells and representing spatial arrangements is with a graph neural network. Lu et al. detected and classified nuclei across each WSI [61]. Each WSI can contain hundreds of thousands of nuclei, so they grouped neighboring nuclei into clusters. Then they formed a graph of cellular architecture by connecting neighboring clusters. They applied a graph convolutional network to predict receptor status of breast cancer, outperforming CNN methods operating on image tiles. In a follow-up study, Lu et al. extended their method to make it more interpretable by producing predictions for each graph node in addition to the overall graph [62]. While fewer studies have modeled cell types, densities, distributions, and spatial arrangements for predicting molecular biomarkers, these approaches clearly present opportunities for future exploration.

### 2.3. Strongly Supervised Biomarkers

For the majority of molecular biomarkers, we only have patient- or slide-level labels. Protein markers assessed with immunohistochemistry (IHC) are an exception. By staining serial sections or restaining a single section and aligning the WSIs, localized ground truth annotations for breast cancer receptor status can be obtained. Gamble et al. used H&E slides co-registered with ER, PR, and HER2 IHC, and pathologists scored 512×512 pixel tiles for each receptor [63]. They trained a model for each receptor on these tiles to predict positive, negative, or non-tumor. Slide-level predictions were made with a model that aggregates tile predictions. Co-registered IHC/H&E images may be difficult to obtain, but, when available, they provide very helpful annotations for predicting receptor status.

The annotations created by Gamble et al. are at the image tile level, however the status of individual cells can also be observed with IHC. Liu et al. created a cell-based deep learning method to predict Ki-67 status [64]. Pathologists annotated positive and negative Ki-67 regions on the IHC slides and labeled each cell within the regions as positive or negative. Then they transferred these labels to H&E and trained a CNN on 64×64 pixel tiles.

As a final example, Su et al. proposed an alternative means that requires no manual annotation [65]. They used the IHC stain p53 as a marker for cancer, enabling molecular label transfer. After registering H&E and IHC images, they transferred p53 positivity to image tiles. Then they trained a CNN to predict the p53 status of each tile. While p53 is correlated with tumor regions, some non-cancer regions stain positive and some cancer regions stain negative. To accommodate this discordance, they excluded p53 negative regions from cancer slides and p53 positive regions from non-cancer slides to create a cleaner training set for cancer detection.

These three examples of molecular label transfer demonstrate the power of strong supervision when it is available. Multiplex immunofluorescence [66] and spatial transcriptomics may provide alternative means for labeling cells or tiles in future research.

## 3. Challenges

While the bottom-up, pathologist-driven, and strongly supervised models discussed above cover the majority of machine learning research to predict molecular biomarkers, there are some important additional considerations for generating robust models. Most frequently, tumor tiles are selected from each WSI, but other tissue selection approaches are possible. Artifacts from slide processing and scanning can obscure relevant biology, making quality control measures essential. Model validation is difficult when large and diverse datasets are not always possible, opening up opportunities for bias, batch effects, and poor model generalizability. Explainability for black box deep learning models continues to be an active area of research. This section will delve a bit deeper into each of these challenges and highlight some novel ideas that have been used to target each.

### 3.1. Tile Selection

Some studies have relied upon manual annotations of tumor regions from pathologists. This could take the form of tissue microarray cores that are selected by pathologists [17,30,39,67] or annotated WSIs [16,23,34,51]. Others have elected to develop a tumor vs. non-tumor model, either to classify each tile or to compute a pixel-level segmentation for the WSI [18,24]. Depending on the model, all tumor tiles may be used to train a biomarker model or some subset. Some studies select a random subset of tiles [43], while others are more strategic. For example, Xu et al. selected representative tiles by clustering all tiles and choosing tiles from each cluster [18].

While most studies include only tumor tissue, others have experimented with including non-tumor tiles. Rawat et al. compared models that used all tissue tiles, epithelium only, stroma only, fat only, or epithelium and stroma for predicting breast cancer receptor status [30]. The best results were achieved using epithelium and stroma but models using all tiles or epithelium only were only 1% behind. Muti et al. also experimented on which regions of tissue were included in models: the whole slide, tumor only, or a virtual biopsy of a 2 mm wide region of tissue [68]. While tumor only performed best, using the whole slide was only slightly behind. Taking the tissue type components idea even further, Campanella et al. used a semantic segmentation model to locate thirteen different subtypes of tissue [49]. They trained tissue-type specific models to compare with a tumor-only model. The best tissue-type specific models improved or were on par with tumor-only models, indicating that some tissue types have fewer discriminative morphologic traits.

The attention component discussed in Section 2.1.2 makes it possible to train with both tumor and non-tumor tiles, allowing the model to identify the discriminative tiles. Schrammen et al. also included slides containing no tumor in their training set [69]. Tissue tiles from non-tumor slides were labeled as non-tumor and tiles in tumor slides were labeled according to their slide label, making this a multiclass model. Their method performed slightly better than a two-class model that uses all tiles (tumor and non-tumor). Their method is also simpler than many of the multiple instance learning strategies commonly employed.

### 3.2. Magnification for Bottom-Up Approaches

An open question for all bottom-up deep learning models is what magnification level should be used? Cellular details are not visible at lower magnifications. Yet, contextual information is reduced at higher magnifications as a smaller region of tissue can be contained in a tile of a fixed pixel size. Many studies downsized image tiles to fit the input size of a particular CNN architecture [24,67,70]. Others selected a lower magnification for computational efficiency. Gamble et al. succeeded with 5× magnification [63], Sirinukunwattana found 5× best in some cases [22], and Xu et al. found 2.5× better than 10× [35]. Yet, Wang et al. compared with higher magnifications and found 5× to generally under-perform [71]. There is not yet a clear consensus on what magnification level should be used for modeling. It may depend on the type of biomarker.

### 3.3. Quality Control

WSIs generally come with some processing artifacts: tissue folds, uneven sectioning, out-of-focus regions, bubbles, etc. Some artifacts can be caught during scanning and be reprocessed, but others may not. All artifact types have been shown to degrade model performance, depending on the severity [72]. Artifacts are more prevalent in some datasets than others. The most frequently studied WSI dataset is TCGA. Campanella et al. studied EGFR prediction for lung cancer [49]. They trained their model on an in-house dataset and tested on TCGA. Performance on TCGA including all cases was poor; they had to exclude about half the slides that contained artifacts to get decent results. TCGA was set up as genomics project, so the challenges with WSIs may not be too much of a surprise. Some form of quality control is often necessary when applying deep learning to WSIs. The same quality checks should be performed on both training and inference data.

Another quality aspect to be aware of is formalin-fixed paraffin-embedded (FFPE) samples versus frozen sections. Misleading artificial structures such as nuclear ice crystals, compression, and cutting artifacts are common in frozen sections. While some research has been done to enable FFPE and frozen sections to be processed by the same model [73], they generally need to modeled independently [36].

### 3.4. Explainability

With these new prediction capabilities for molecular properties comes the need for explainability techniques to breed trust, ensure safety, enable a path to better models, and generate new biological insights. Seegerer et al. explored interpretable methods for predicting ER status from breast tumor images [74]. They used Layer-wise Relevance Propagation to locate relevant parts of the images for pathologists to examine. Stroma was found important for indicating ER+, while lymphocyte infiltration and high grade were associated with ER-. A larger study will be needed to further validate these observations. Layer-wise Relevance Propagation is just one of many pixel attribution methods. GradCam is also frequently used to produce this type of heatmap [75]. Alternatively, classifier predictions on smaller tiles can be used to generate a class heatmap [30,41,63].

The attention models discussed in Section 2.1.2 bring another opportunity for generating heatmaps. Graziani et al. used attention models to study the spatial distribution of attention weights for different gene expression levels for colorectal cancer [48]. To get a more accurate measure of attention, they averaged attention weights from multiple models and displayed them for each gene as a heatmap over the image. Attention scores combined with the discrimination layer can help pathologists identify morphological associations for each class [49]. In Campanella et al.’s study, pathologists were also able to note common characteristics of false positive and false negative tiles that could lead to strategies for improving model results in the future.

The obvious path to more explainable models is to use interpretable, pathologist-driven features like those outlined in Section 2.2. When training deep learning models, these pathologist-driven characteristics could be generated afterwards as a means of explainability. Xu et al. first trained a deep learning model to predict chromosomal instability [35]. Then they applied the nuclei segmentation and classification model HoVer-Net [76] to learn more about the tiles predicted to have high or low chromosomal instability and found that high regions had larger neoplastic cell nuclei [35]. While interpreting deep learning models remains an active area of research, hybrid solutions like this provide an additional opportunity for opening the black box, possibly leading to learning opportunities for pathologists and providing insights into disease mechanisms.

### 3.5. Validation

Deep learning models are prone to overfitting due to the millions of parameters within them. Models should always be trained and tested on different sets of data. When hyperparameter turning is also performed, a distinct validation set is also necessary: optimize the model parameters on the training set, tune the hyperparameters on the validation set, and measure model performance on the test set. When a dataset is too small for division into these three subsets, cross-validation methods like k-fold or Monte Carlo may be used to produce multiple train/validation/test splits. As will be discussed in Section 3.7, models should never be trained and validated on patients from the same medical center.

Weakly supervised models can be validated at the patient- or slide-level, but it is much more difficult to validate the heatmaps they produce with localized predictions. Protein biomarkers are a unique case because IHC slides of the same or adjacent tissue can be produced. Even if the model is trained with patient-level annotations, IHC can provide a means of qualitative validation. Shamai et al. explored this for breast cancer ER status [67]. Spatial transcriptomics, where the spatial distribution of mRNA expression is measured across a tissue section, can provide another means for validation [77].

Validation is also important for understanding model performance and identifying categories in which model predictions are more likely to be incorrect. Echle et al. categorized causes of incorrect predictions from their MSI model on colorectal cancer [70]. While uncommon or challenging tissue components were associated with false positives, artifacts, small amount of tumor, or not tumor were common causes of false negatives. Naik et al. calculated model performance for a number of different stratifications of histological and clinical variables [78]. Some affected model performance significantly more than others. This method of analyzing model performance is critical to understanding how well it generalizes and where improvements could be made in the future. It can also identify hidden batch effects [79]. Zeng et al. went as far as validating other methods of gene expression profiling, staining protocols, and WSI formats [51]. Validation on external cohorts of data is essential. Javed et al. outlined some key mechanisms for validating models for pathology images [80].

### 3.6. Domain Generalizability

Some of the studies discussed in this article assess performance on an independent cohort, while others simply split a single dataset into training, validation, and test sets. Those that do test on an independent cohort typically see a performance drop because the test set was collected at a different hospital, processed in a different lab, imaged with a different scanner, or represents a different population of patients [36,43]. Generalizability to all of these variations may not be needed for every application. For example, if tissue is processed in a central lab with a single scanner manufacturer and a consistent staining procedure, then fewer sources of variation must be accommodated. The needs for your application should be defined and the limitations of your model validated and understood.

Sirinukunwattana et al. took steps to improve generalization performance by applying adversarial domain training to encourage their model to learn domain-invariant features [22]. Alternatively, Rawat et al. accommodated variations in staining by different labs using a GAN-based style transfer [30]. A large portion of the improvement in their model was attributed to this step. In yet another approach, Yamashita et al. used augmentation to improve out-of-domain performance by replacing the style of the image (texture, color, and contrast) with the style of a randomly selected image while preserving the semantic content [81]. Self-supervised pretraining has also been shown to produce more generalizable models [41,44]. A full assessment of domain adaptation and generalization approaches is beyond the scope of this article, but these adversarial domain training, stain normalization, augmentation, and self-supervised techniques can make significant strides towards generalizability.

### 3.7. Batch Effects

The vast majority of studies discussed in this paper use TCGA due to its easy availability. Regardless of the dataset selected, one should always evaluate the characteristics of the training dataset as it may affect how a model performs on different test sets. Howard et al. found batch effects across multiple cancer types in TCGA, resulting in biased estimates of performance when they were not controlled for [82]. Stain normalization can remove some of the variations and augmentation can mask differences in color, but second order image features remained and enabled a deep learning model to identify the tissue submitting site. Howard et al. advocated for reporting variations in model performance across sites and, ideally, never training and validating on patients from the same site.

Further evidence of this batch effect was provided by Dehkharghanian et al. when they trained models on TCGA [83]. The models were not intentionally tuned to predict acquisition site—and yet they could. At first glance, this may not be a concern because different sites have different staining techniques and scanners. However, the important factor to consider is that the distribution of clinical information such as survival and gene expression patterns also significantly differs from site to site. Biased models might be identifying the institution that submitted a sample rather than predicting prognosis or mutation status. Dehkharghanian et al.’s advice was similar to Howard et al.: stratify training and test sets by the tissue source site to prevent the model inadvertently learning features of the site.

Tissue microarrays (TMA) present additional opportunities for batch effects and bias in comparison to WSIs due to multiple patient samples being on the same slide. Bustos et al. created a bias-ablated deep learning solution to reduce model bias to study project, patient, and TMA glass [84]. Their method works similarly to domain adversarial training in that they optimize their model for the desired task while also reducing correlation between model activations and batch effects. This additional objective helped the model avoid learning the undesirable batch effects.

### 3.8. Dataset Diversity and Spurious Correlations

Tissue source site is not the only potential source of bias and degraded model performance. Patient population is another key factor. Different cohorts of patients may have a different distribution of age, gender, race, or some other subgrouping. The types of distribution shifts that can degrade model performance are generally understood (spurious correlation, low data drift, and unseen data shift), but the most suitable solution is inconsistent across datasets [85,86,87].

As an example, Ektefaie et al. looked for spurious correlations between biomarkers in their data [88]. They verified whether the model was still accurate on samples that were not concordant. They performed this analysis for ER and PR, as well as ER and lymphocytes. In the case of the former, they found that that the model was still fairly accurate when ER and PR were discordant. For the latter, the ER classifier learned morphology resembling lymphocytes to aid in distinguishing ER+ from ER-. The first step is to detect spurious correlations. The second step is deciding how to handle them.

One solution was outlined by Lazard et al. in their working on predicting homologous recombination deficiency (HRD) in breast cancer [89]. Initially they developed their model using the TCGA dataset. However, they found that the molecular subtype was a significant confounder. Upon correcting for this bias, performance dropped significantly, indicating that their model for HRD prediction was predicting to some extent the molecular subtype. So they pivoted to developing subtype-specific models for luminal and triple negative breast cancer on an in-house dataset that is more homogeneous. They used a strategic sampling strategy to balance their training data with respect to the output variable and to correct for bias. Because of their use of a homogeneous dataset, their methods require further validation on multi-center cohorts. However, the insights revealed from this study provide critical information on mitigating bias.

Sometimes the best solution is creating a more diverse training set. Muti et al. experimented with ten cohorts of patients to detect Epstein-Barr and MSI in gastric cancer, first training a model on each cohort individually [68]. Performance varied significantly across the cohorts. Then they trained a model across multiple cohorts and tested it against a held out set of cohorts, and performance was much better.

### 3.9. Small Datasets

WSIs contain billions of pixels. In fact, fewer than fifty WSIs will provide more pixels of tissue than the 1.2 million photos in ImageNet. However, it is the small number of patient samples that causes challenges. Unlike computer vision benchmark datasets that get larger each year, medical applications are often limited by the number of patients because of the time and expense of processing patient samples, the limited number of patients with a particular disease, and privacy concerns. What qualifies as a small dataset will generally depend on the diversity of the dataset and the complexity of the prediction task.

When only a small number of patient samples is available for training, weakly supervised learning is much more challenging. The strongly supervised methods discussed in Section 2.3 are one possible solution. The transfer and self-supervised learning approaches discussed in Section 2.1.1 are a common approach to enable training a CNN on a larger unlabeled or weakly labeled dataset. Image tiles can then be encoded with the pretrained network and a simpler classification model applied to produce a final prediction. Standard regularization techniques like image augmentation can also be used to increase the diversity of the training data.

Another solution is through image augmentation. generating additional synthetic tiles. Krause et al. trained a Conditional Generative Adversarial Network on their training set of colorectal cancer images to create new sample images with and without MSI [90]. Augmenting their training set with these synthetic images increased the size of their training set and improved model accuracy.

Privacy concerns can be the limiting factor in creating larger datasets as medical centers may be prohibited from sharing sensitive patient data. Federated learning has emerged a new paradigm for training on decentralized datasets while preserving privacy, enabling models trained on data from multiple medical centers [91]. In an alternative called swarm learning, there is no central location for computation, so the model is trained jointly across different partners. Saldanha et al. trained models for BRAF and MSI prediction using three training cohorts and two validation cohorts [92]. Swarm learning consistently outperformed models trained on a single cohort and was on par with multi-cohort models, especially when the training cohorts were weighted according to the number of patients they each had. If multi-cohort models are not possible and your dataset is relatively small or lacks diversity, swarm learning can provide a great alternative by sharing model weights but not patient data.

## 4. Opportunities

Despite the considerable challenges with biomarker prediction models, research continues to reveal new opportunities and use cases. Models can enable the quantification of intratumoral heterogeneity. Predicted biomarkers have been associated with patient outcomes. Some studies have looked more broadly across cancer and biomarker types to demonstrate a single framework that works for all and enables researchers to study what types of biomarkers are most successfully predicted from histopathology. New model types are emerging in the rapidly advancing computer vision and deep learning research that have yet to tested on biomarker prediction. Publicly accessible datasets and code bases make it feasible for anyone to get started easily. This section will examine each of these opportunities in more detail.

### 4.1. Biomarker Heterogeneity and Outcomes

Localized predictions, such as with the heatmaps discussed in Section 3.4, provide an additional benefit in understanding tissue heterogeneity. For example, Levy-Jurgenson et al. developed a model to predict mRNA and miRNA expression from WSIs [55]. They referred to the heatmaps produced as molecular cartography and used them to study heterogeneity. Examining a small set of genes at a time, they displayed the model predictions over the slide, coloring each region based on the predictions. To quantify heterogeneity, they computed the fraction of the slide belonging to each group and applied Shannon’s entropy formula. High heterogeneity was found to be linked with poor survival, especially for breast cancer.

Intratumoral heterogeneity is not the only output associated with patient outcomes. Jaber et al. used deep learning to predict Basal versus Luminal A genomic subtypes of breast cancer [93]. The image-based biomarkers were found to be more predictive of patient survival than the molecular subtypes themselves. Further, the heterogeneous samples-predicted to contain both Basal and Luminal A subtypes-had an intermediate prognosis. Bychkov et al. also demonstrated that their image-based model can predict patient outcomes [75]. They trained their model to predict breast cancer HER2 status and demonstrated that these predictions were also associated with survival and treatment response. While these biomarker models were not trained on patient outcome or treatment response directly, the biomarker predictions may provide additional information for treatment planning.

### 4.2. Pan-Cancer Modeling

Most of the studies discussed in this article focus on a single or small set of cancer types and molecular properties. A few have experimented more broadly across cancer types. Kather et al. tested their algorithm on WSIs from 5000 patients for 14 different tumor types [26]. They created a single deep learning workflow to predict point mutations, molecular and genomic subtypes, and hormone receptor status from H&E images. Not every molecular property could be accurately predicted for every cancer type, but a large portion of them were shown to be significant. Fu et al. performed a similar test for predicting genomic, transcriptomic, and survival properties across 28 cancer types [33]. They used a relatively simple setup with transfer learning to extract a feature vector from each image tile. Then a linear classifier used the feature vectors to predict the class of each image tile, with the slide-level prediction computed as the average across tiles. Even in this simpler setup without tuning the CNN for histopathology, they found links between tumor morphology and molecular composition in every cancer type and for almost every class of genomic and transcriptomic alteration. Arslan et al. trained 13,443 deep learning models to predict 4481 different biomarkers across 32 cancer types [94]. Through this extensive analysis, they were able to assess which types of biomarkers are most accurately predicted from H&E. Standard of care biomarkers performed best, followed by clinical outcomes and treatment response, protein expression, gene signatures and subtypes, gene expression, driver SNV mutations, and metabolomic pathways.

While the above examples developed a single framework for modeling many types of biomarkers and cancers, others have experimented with training a model on a single type of cancer and testing it on another type. Qu et al. trained models on breast cancer and tested them on lung and liver cancer [19]. The models did not perform as well as on breast cancer, although the mutation results were better than those for biological pathways. On the other hand, Jang et al. tried applying colorectal models for mutation status to gastric cancer and they failed to generalize [36]. Different cancer types have different morphology, so this is not too surprising.

### 4.3. Multimodal Models

In a parallel line of research, molecular biomarkers are also predictable from radiology images, including CT, PET, or MR [95,96,97,98]. Some of these works rely on hand-crafted features and classical machine learning, while others use CNNs to learn relevant features. Other studies have gone a step further to employ multiple modalities of data in predicting molecular biomarkers [99,100]. Future research may expand the power of multimodal learning to predict biomarkers, enabling more accurate models that take advantage of clinical and omic data.

### 4.4. New Model Types for WSIs

Weak supervision on gigapixel WSIs generally requires splitting into tiles in order to fit on a GPU. Some recent techniques now enable a WSI to be processed as if it were all held in GPU memory. A CUDA feature called unified memory provides the GPU direct access to host memory. Similar to virtual memory, pages are swapped on the GPU when requested. Through this technique, Chen et al. were able to process images as large as 20 k × 20 k pixels [101]. Any larger became prohibitively slow. To accommodate larger WSI, they downsized them by a factor of four in each dimension. This technique is best suited for lower magnification images.

Another alternative is a method called streaming that exploits the locality of most CNN operations. It combines precise tiling and gradient checkpointing to reduce memory requirements. To stream the forward pass of a CNN, you first calculate the feature map of a chosen layer in the middle of the network. This layer is smaller than the original image because of downsampling, so it fits on the GPU. This reconstruction of the intermediate feature map is then fed to the remainder of the network. The backward pass is computed in a similar fashion. Pinckaers et al. first demonstrated this technique with a small and simple CNN on 8 k × 8 k images [102]. They subsequently showed it for a ResNet on 16 k × 16 k images [103]. The limitation of streaming is that it cannot handle feature map-wide operations such as batch normalization in the streaming (lower) part of the network. As a workaround, they froze the mean and variance of batch normalization layers.

Yet another possibility is a spatial partitioning approach called halo exchange [104]. Hou et al. demonstrated this for segmentation of 512×512×512 CT images and speculated that it would also be a good fit for histopathology. This technique distributes the input and output of convolutional layers across GPUs with the devices exchanging data before convolutional operations.

From unified memory to streaming to halo exchange, each of these approaches enables end-to-end training of much larger images—but still with current limits around 20 k × 20 k pixels or less. With future advances, these types of techniques may enable us to train end-to-end models on WSIs at 20×. These new classes of models could provide a means to better capture tissue architecture in addition to local morphology, perhaps leading to more accurate predictions.

Another new class of models suitable for weak supervision is transformers. If self-supervised learning on image tiles is used to downsize the WSI representation, then it becomes feasible to apply a transformer as an MIL model to capture the interaction between instances. While there are no studies yet that use transformers for MIL in predicting biomarkers, they have been applied to detecting tumor and metastases, classifying histologic subtype, and predicting patient outcomes [105,106,107,108].

### 4.5. Datasets and Challenges

The most common histopathology dataset used for computational pathology research is TCGA. With more than 20,000 samples across 33 different cancer types and genomic information for each, TCGA has created the opportunity to study biomarker prediction not possible previously. It also presents many of the data challenges discussed in Section 3. Further, different studies on the same biomarker and the same cancer type frequently make different decisions about which patients to include and how to create their training, validation, and test sets. Processing the WSIs and selecting which tiles to include creates even more diversity in the data used. This makes it quite challenging to compare results from different studies.

There are, however, some benchmark datasets. Kather released two different versions of the colorectal MSI dataset used in some of his research [109,110]. Both are from TCGA, the WSIs have been tiled, and non-tumor tiles were discarded. Stain normalization was applied to one of the datasets [109]. The other benchmark dataset is HEROHE, which was a challenge at the European Congress on Digital Pathology in 2020 [111,112]. The goal was to predict HER2 status from H&E in invasive breast cancer. Both datasets have resulted in numerous publications.

## 5. Discussion

This article has reviewed numerous types of machine learning models for predicting molecular biomarkers from H&E. While pathologist-driven methodologies are used in a minority of studies, they do bring an easier path to model explainability—something that is still a challenge for deep learning. The cell map and graph neural network approaches based on detecting and classifying nuclei can provide a powerful and somewhat interpretable means to more fully characterize the tumor microenvironment. Both types of pathologist-driven models are based on a pathologist’s view of important tissue characteristics. This can lead to a limited power to distinguish tissue classes that are too complex for a pathologist to identify visually. The best alternative is bottom-up approaches like deep learning. Especially when combined with recent advancements in self-supervised learning and weakly supervised attention, these models enable learning local features and allow the model to identify which tiles are discriminative. Self-supervision is most beneficial when your data are significantly different from standard benchmark datasets like ImageNet, and you have a large amount of unlabeled or weakly labeled data from which to learn a representation. By pretraining with self-supervised learning, the model can compress image tiles before applying an attention model to the whole slide. This approach has shown great results and even seems to generalize to new datasets better than alternative models. If strong supervision is possible—for example, using aligned H&E and IHC—this creates a means for molecular label transfer in which a biomarker could be trained on a smaller dataset due to the stronger labels.

There remain a number of challenges in developing machine learning models for predicting biomarkers. Section 3 detailed them to the extent they’ve been explored. However, the following questions still remain:
Which regions of tissue should be included when training a model?What magnification is best?What effect do artifacts have? Can they be detected and discarded systematically?Can we create more explainable models?How thorough do we need to be in validating with external cohorts?What batch effects do we need to watch out for?How can we be sure that we’ve detected all spurious correlations?Can we create models that generalize to different scanners, medical centers, or patient populations?How do we mitigate bias?Can we train models with a small number of patient samples?

There may not be a single answer to these questions. It may depend on the characteristics of the dataset you’re working with. However, methodologies for exploring and answering these questions are still needed.

Despite these challenges, successful insights have already been revealed. Insights into intratumoral heterogeneity and associations with patient outcomes demonstrate the powerful capability of these methods. A single framework can be used to explore many biomarkers and many cancer types. New model architectures may make it possible to train end-to-end in a way that better models tissue architecture, and transformers present a new type of MIL model. Benchmark datasets have been created to enable a consistent comparison of algorithms.

Deep learning-based biomarkers from H&E could provide an additional tool for pathologists and new insights for companion diagnostics and drug discovery [3]. Based on only H&E images and no longer limited by the amount of tissue excised or the processing time for individual molecular tests, deep learning-based methods can provide easier access to vital information about a tumor. They may yield a screening method to determine when to run molecular tests or an alternative when molecular tests are not possible.

## Figures and Tables

**Figure 1 jpm-12-02022-f001:**
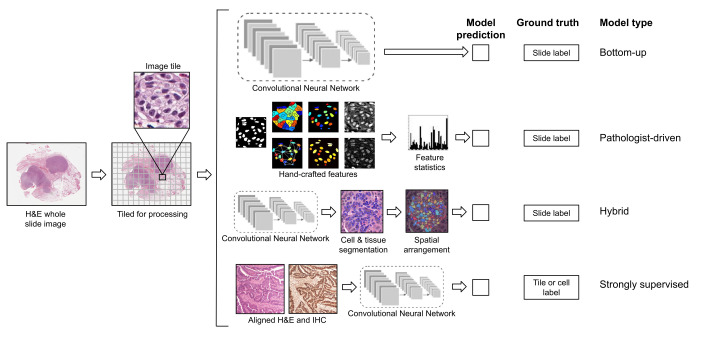
Workflow and general frameworks for predicting molecular biomarkers from H&E: (1) bottom-up methods that learn patterns directly from the images, (2) pathologist-driven features, (3) hybrid workflows, (4) strongly supervised models.

**Figure 2 jpm-12-02022-f002:**
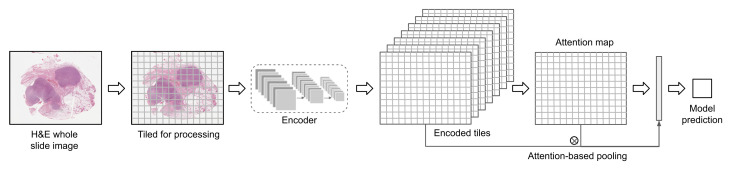
Multiple instance learning with an attention model: (1) WSIs are split into tiles for processing, (2) tiles are encoded with a CNN or some other model, (3) an attention map is calculated from the tile features, (4) tile features are aggregated using the attention weights, (5) a final class is predicted.

## Data Availability

Not applicable.

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
