# Peer review of "Deep Learning-Based Prediction of Molecular Tumor Biomarkers from H&E: A Practical Review"

_jpm, 2022, doi:10.3390/jpm12122022_

Round 1
Reviewer 1 Report
JPM-1976166 Deep Learning-Based Prediction of Molecular Tumor Biomarkers from H&E: A Practical Review
This review summarizes the current state of the art for prediction of molecular markers from image data alone, without the need for a physical assessment of a biopsied tissue sample. Such a procedure allows for stratification of patients based on their likelihood of responding to a certain kind of therapy, such that they receive the most personalized care for their own personal and disease specifics. This review touches on many recent and important developments in this space, and would be of interest to the readership of JPM. There are, however, suggestions that I will make to potentially improve the article prior to further consideration for publication.
[1] Could the author describe some different methods for cross-validation to assess model training? I envision perhaps mentioning k-fold and Monte Carlo cross-validation techniques.
[2] In the context of slide images, is data augmentation through the use of rotations/scaling/shearing as effective as augmenting images on the patient-level? Is there any work on the slide-level that has used this kind of augmentation, or is it typically replacing texture/color/contrast without touching the spatial features of the slides?
[3] Has any work been performed to make these molecular marker predictions without even needing the tissue biopsy? Essentially using the external imaging already performed as part of the standard workup (CT, MRI, etc) and make this stratification.
[4] Following up on the previous point, how might radiomics play a role in this problem space? Perhaps discussion on leveraging the image markers and ensembling them with the biomarkers and perhaps other clinical data like patient demographics would be a useful inclusion into this review article.
[5] When we use the term "small dataset", what does that really mean? Is it task dependent or dependent on the type of data you are using to train the algorithm? Are there specific deep-learning methods that have been developed to better deal with a dataset of limited size, beyond augmentation?
[6] How generalizable do models have to be? Does it make sense to develop a local algorithm robust for image data or clinical practice of an outside institution? Could discussion be added regarding how to achieve the appropriate amount of generalizability?
[7] Your formatting of the citations in the text didn’t quite translate over to what was uploaded for review, so it is difficult to assess exactly what works were being cited and at which points in the text.
Author Response
Dear Reviewer,
Thank you for your valuable feedback.
You’ll see my revisions in red text in the updated manuscript pdf. Responses to your specific comments are below.
[1] See new paragraph in section 3.5.
[2] I have only seen augmentation done at the image patch level. If you have a reference that performs augmentation at the slide level, I’d love to read it.
The only other related technique I can think of is if multiple slides of the same tumor are available. Perhaps some are not H&E but through virtual staining could be transformed into H&E and used as additional training examples. But I can’t think of anywhere I’ve seen this in the literature. Please share a reference if you have.
[3+4] You asked about “using external imaging already performed as part of the standard workup.” While histopathology is not “external imaging” it is the gold standard for diagnosis of cancer. Other types of imaging may also enable patient stratification, but they won’t completely replace the need for tissue biopsy any time soon.
I added a paragraph about predicting biomarkers from radiology images and multimodal data. See section 4.3.
[5] Section 3.9 addresses small datasets, including suitable DL approaches. I have clarified the term “small dataset.”
[6] Good point. Generalizability needs are application dependent. I expanded on this in section 3.6.
[7] I submitted the manuscript in Latex, so I’m also confused about why the citations didn’t show up in the compiled pdf. I will follow up with the editor about this.
Reviewer 2 Report
The Author presents a very well written review on the topic of prediction of biomarkers on H&E. It is structured and written clearly, from an expert point of view, with appropriate references and sound conclusions.
A few observations:
Some of the work appears to have already been published by the same author here https://pixelscientia.com/articles/predicting-molecular-tumor-biomarkers-from-he/ . I am not sure if this constitutes plagiarism, but if it does, given the quality of this manuscript, I would suggest revising it to fix the issue.
Another problem: all references are not evaluable since they appear as [???] and are not present at the end of the manuscript.
Finally, the paper is quite long and has a long introduction before settling on the topic of biomarker prediction. Iff the editor thinks it is necessary, I feel it could be shortened a bit without losing too much meaning.
Minor points:
L18. Wrong from a medical point of view. It should be "tumors that overexpress" (not "test positive", since weakly positive tumors are not amenable to anti-HER2 therapy)
furthermore, anti-HER2 therapy such as herceptin (trastuzumab) is technically not immunotherapy, but simply targeted therapy (it is an antibody indeed, but immunotherapy is a different thing)
L29 The first two paragraphs of the introduction contain numeorus repetitions. They might be fused in a single paragraph without sacrificing meaning.
L41 overstatement. Perhaps new slices of a FFPE block are required, but in the overwhelming majority of cases, a new tissue sample is not required.
L284: p53 can be negative in cancer and that constitutes a pathologic (i.e. positive) result. I cannot evaluate whether this misstatement is present in the quoted study as well since the reference is not visible.
L477 do you need the TNBC abbreviation if it is never used elsewhere?
Author Response
Dear Reviewer,
Thank you for your valuable feedback.
As you noticed, I posted an earlier version of this article on my website. The editor has confirmed that preprints are allowed and recommended that I reference an article that I wrote for the Digital Pathology Association [3].
I submitted the manuscript in Latex, so I’m also confused about why the citations didn’t show up in the compiled pdf. I will follow up with the editor about this.
You’ll see my revisions in red text in the updated manuscript pdf. Responses to your specific comments are below.
Introduction has been shortened.
L18: Corrected – thank you!
L29: Reworked the intro for clarity and to reduce repetition.
L41: Corrected – thank you!
L284: This is the reference about p53:
Su, A.; Lee, H.; Tan, X.; Suarez, C.J.; Andor, N.; Nguyen, Q.; Ji, H.P. A deep learning model for molecular label transfer that enables cancer cell identification from histopathology images. NPJ precision oncology 2022, 6, 1–11.
This study does acknowledge that p53 can be negative in cancer.
I believe my description addresses this: “While p53 is correlated with tumor regions, some non-cancer regions stain positive and some cancer regions stain negative. To accommodate this discordance, they excluded p53 negative regions from cancer slides and p53 positive regions from non-cancer slides.”
However, I did add “to create a cleaner training set for cancer detection” to the end of the above sentence to make it clearer.
Please let me know if there’s a connection I’m still missing.
L477: Removed TNBC abbreviation.
Round 2
Reviewer 2 Report
All changes are satisfactory. Congratulations to the Author for such fine contribution which will be a pleasure to re-read.